# Amoxicillin-Induced Atypical Exanthema in a Patient with EBV-Related Nasopharyngeal Carcinoma: A Case Report

**DOI:** 10.3390/v17030368

**Published:** 2025-03-04

**Authors:** Matteo Carpani, Davide Smussi, Andrea Esposito, Francesca Consoli, Alfredo Berruti, Andrea Alberti

**Affiliations:** Medical Oncology Unit, Department of Medical and Surgical Specialties, Radiological Sciences and Public Health, University of Brescia, ASST Spedali Civili, 25123 Brescia, Italy; m.carpani001@studenti.unibs.it (M.C.); davidesmussi@gmail.com (D.S.); a.alberti015@unibs.it (A.A.)

**Keywords:** nasopharyngeal carcinoma, EBV, amoxicillin, antibiotics, skin rash, case report

## Abstract

Introduction: The concomitant use of antibiotics, especially beta lactams, during acute EBV infection is widely associated with an increased risk of skin manifestations; the actual pathophysiology and prevalence of this phenomenon are still debated. Case report: We present the first reported case of atypical exanthema associated with amoxicillin intake in a patient with EBV-related nasopharyngeal carcinoma. We recorded a pattern in the plasma EBV-DNA load consisting of a significant increase at the onset of the rash with a sudden remission after its resolution. The patient recovered without sequelae. Discussion/Conclusions: The temporal relationship and the reported data on rash morphology, clinical findings and triggering factors support a possible correlation between the intake of beta-lactam antibiotics, particularly amoxicillin, and the onset of cutaneous manifestations in a patient with nasopharyngeal carcinoma. Such reactions can be a challenging differential diagnosis and may warrant increased provider consideration when choosing to prescribe beta lactams in patients affected by nasopharyngeal cancer.

## 1. Introduction

Amoxicillin is among the antibiotics that most frequently cause cutaneous toxicity, with an estimated incidence of 1% in exposed patients. The main manifestations are itching, erythema, various types of skin rash and urticaria, and, in severe cases, manifestations such as Stevens–Johnson syndrome [1]. Skin manifestations are a relatively common occurrence during acute EBV infection, especially following the concomitant intake of antibiotics such as ampicillin and amoxicillin but also azithromycin, levofloxacin, piperacillin–tazobactam and cephalexin [2]. The pathophysiology is still doubtful, and the most established theory implies a loss of tolerance induced by the viral infection, which leads to transient Th1-mediated type IV hypersensitivity; however, a role both of molecular mimicry between the viral proteins and the amoxicillin molecule and of a transient failure of plasma detoxification mechanisms, particularly a lowering of glutathione levels, cannot be excluded [3]. The actual prevalence of the phenomenon is still debated; historical case series estimate the occurrence of rash in approximately 70–90% patients with acute EBV infection treated with antibiotics [4], while more recent ones downgrade the rate to 18–30% [5,6].

## 2. Case Presentation

Z.E.C. is a female patient who presented to our unit in August 2019 with a non-keratinizing EBV-related nasopharyngeal carcinoma, occupying the Rosenmüller fossa, and with a latero-cervical adenopathy in the left II level of the neck. The patient underwent radiotherapeutic treatment (RT) with curative intent from October to November 2019 on the nasopharynx and neck at a dose of 69.3 Gy in 33 fractions, administered with concurrent chemotherapy based on weekly cisplatin (40 mg/m^2^). Following a 17-month disease-free interval (DFI), the patient developed a pleural lesion at the level of the left fifth rib, and the EBV DNA load increased from undetectable to 1006 copies/mL. The patient was then treated with a first-line systemic therapy with carboplatin and gemcitabine from June 2021 to October 2021, followed by maintenance gemcitabine until January 2022; during the treatment, EBV DNA progressively reduced to undetectable values (<94 copies/mL). As the pleural lesions progressed, EBV DNA peaked to 754 copies/mL, and the patient was then enrolled for a second-line systemic treatment in the POINT trial (NCT04825990), a single-arm, phase II clinical trial that aimed to assess the activity of a combination of pembrolizumab and olaparib in patients with recurrent/metastatic, platinum-resistant nasopharyngeal carcinoma [7,8]. The patient had a partial response as the best response and received pembrolizumab + olaparib from April 2022 to January 2024.

In the October 2023 restaging CT scan, an oro-nasal fistula and a 32 × 18 mm consolidative, irregular and partially excavated lesion in the right superior lobe of the lung were reported, with the latter being interpreted as an infective lesion after radiologic revision in the Multidisciplinary Meeting. Based on the previously described findings, the oncological therapy was temporarily suspended and an antibiotic treatment with amoxicillin–clavulanic acid and azithromycin was administered from the 31st of October to the 4th of November.

On the 6th of November, the patient contacted our unit reporting the onset of an erythematous skin rash, which was initially localized at the level of the trunk and with a geographical appearance. In the following days the rash extended to the face and to upper and lower limbs, assuming a more compact and widespread distribution with the fusion of the skin lesions described above. The patient also reported generalized pruritus, denying known drug allergies, fever or other symptoms. The patient had previously been treated with amoxicillin before cancer diagnosis, without developing any cutaneous reaction. The physical examination revealed diffuse maculopapular erythematous rash on the trunk, the extensor surface of upper limbs, the proximal portion of lower limbs and the face, finely scaling and modifiable with acupressure, in absence of calor and signs of edema (Figure 1). No other pathological findings were underlined at neck, thoracic and abdominal examination. Blood tests, performed on the same day, showed neither leucocytosis nor hyper eosinophilia, clinical chemistry was normal (except for G1 hypoalbuminemia), and C-reactive protein (CRP) was slightly increased (5.7 mg/dL, ULN 5 mg/dL). The plasma EBV-DNA load was 4006 copies/mL, significantly increased in respect to 4 weeks before (250 copies/mL).

## 3. Outcome

The clinical findings were suggestive of an atypical exanthema, for which dermatological and allergological consultations were requested. The dermatologist described the rash as diffuse, maculopapular and erythematous, implying a possible drug-related etiology; the allergologist, also suspecting the same etiology, recommended avoiding new amoxicillin administrations in the future.

The suspension of the oncological therapy was maintained, and, according to allergological indication, antihistaminic therapy (loratadine 10 mg q.d.), systemic steroid therapy (prednisone 25 mg q.d.) and topical steroid therapy (propionate clobetasol 0.05% t.i.d) were therefore administered. At the following oncological follow-up visit, held on 1 December 2023, the previously described skin rash had completely resolved without sequelae and the plasma EBV-DNA load had returned to the value of 255 copies/mL (as shown in Figure 2).

The patient remained off therapy until March 2024 due to the persistence of pneumonia, and concurrently, a progressive increase in the EBV-DNA load was documented (as shown in Figure 2).

The restaging CT scan in March 2024 showed dimensional progression of the right fifth rib lesion; therefore, third-line systemic therapy with paclitaxel q21 days (175 mg/m^2^) was administered.

The patient is currently still alive and is continuing systemic treatment.

## 4. Discussion

To the best of our knowledge, no cases of antibiotic-related skin rash in patients with EBV-related nasopharyngeal carcinoma have been reported in the literature. However, reviewing the available evidence, the presented case shows several analogies with cases of antibiotic-related skin rash reported in pediatric patients with infectious mononucleosis in terms of the onset of symptoms, morphology of skin manifestations, associated symptoms, type of antibiotic administered, and changes in the EBV titer.

As far as timing is concerned, skin manifestations appeared two days after the end of the antibiotic treatment, and seven days after its beginning. This is consistent with what has been reported in the literature for antibiotic-related skin rash during acute EBV infection, where the onset of the rash is described within an interval of 7–10 days after the start of the antibiotic therapy [9].

Morphologically, scarlet erythematous macular rash has mainly been described in cases reported in the literature, arising on the extensor surface of the limbs, with subsequent extension to the trunk and extremities. The presented case shows several similarities in morphology to what has been reported, being itself an erythematous maculopapular rash with onset on the extensor surfaces; however, onset on the trunk, the involvement of the face and the non-involvement of the extremities, especially the lower limbs, are less frequently reported [2,3,9,10].

According to various reports, this particular type of rash may be clinically asymptomatic or may present with generalized pruritus, fever or malaise and, in rare cases, also with severe presentations mimicking DRESS syndrome. The case considered is consistent with what has been reported, presenting with only the symptom of generalized pruritus [3,9,10].

These skin manifestations are reported to be mainly related to the intake of amoxicillin and ampicillin, with some cases of rash also related to levofloxacin, piperacillin–tazobactam, cephalexin and azithromycin. What is reported in this case is consistent with the current evidence as the patient was treated with amoxicillin–clavulanic acid and azithromycin [2,3]. The rash could not be related to a drug hypersensitivity as the patient had no known al-lergies in her medical history and had been treated with amoxicillin before cancer diagnosis without any reaction.

Another significant finding is represented by the EBV-DNA trend: in this case report, a spike in the EBV-DNA titer can be observed at the occurrence of the skin rash, remitting to low values after its resolution. There is limited evidence concerning the pattern of this marker in antibiotic-related skin rash during acute EBV infection, as EBV-DNA determination is not a standard procedure in the management of infectious mononucleosis. However, Saito-Katsuragi et al. describe the case of an adult patient with an acute EBV infection (initially diagnosed as Still’s disease) who developed a maculopapular rash with similar morphological and temporal features to the one described in this case report after the administration of intravenous (IV) ampicillin; rechallenges with IV ampicillin were performed 39 days, 90 days and 165 days after the acute infection, resulting in the development of a new rash after each administration. The EBV-DNA titer was assayed concurrently and observed to rise 24–48 h before the onset of the rash, settling then to negative values after its resolution. At day 0, day 39 and day 90, the patient was also treated with oral prednisolone (40 mg/day), ruling out the hypothesis of an allergic reaction to the antibiotic [11].

Lastly, different causes of this skin rash seem very unlikely. An immune-related cutaneous toxicity may be excluded, since the presentation was atypical, the administration of pembrolizumab to the patient was suspended at the beginning of October 2023 and, moreover, according to a systematic review by Belum et al. [12], the occurrence of grade 3 or higher rash in patients treated with pembrolizumab is a relatively rare event (incidence 1.7% CI: 0.4–6.7%) and the onset is usually in the initial weeks of treatment (6–20 weeks from the first administration), while in our case, the rash was reported about 2 years after the start of treatment. A cutaneous adverse effect related to olaparib should also be excluded, since this type of toxicity has never been described with this drug [13]. Treatment-related immunosuppression or a deficiency in B- and T-lymphocyte functionality can be ruled out as it is not reported as one of the adverse effects of such treatment; concurrently, disease related immunosuppression can be ruled out as there is no evidence that EBV-related nasopharyngeal carcinoma can affect the functionality of the immune system [14]. Another alternative hypothesis may be represented by a cross-reaction between antibiotic therapy and immunotherapy, but in the current literature, no similar case has been described. In addition, the inappropriate administration of the drug can also be reasonably ruled out; the patient consumed the prescribed antibiotics at the correct dose and for an appropriate length of time. It was also not possible to assess the blood concentration of amoxicillin, as this is not a standard procedure in clinical practice for nasopharyngeal carcinoma. Therefore, we believe that amoxicillin-induced EBV-related exanthema represents the most likely etiopathogenesis of the skin manifestations observed.

## 5. Conclusions

The reported clinical case suggests a possible correlation between the intake of beta-lactam antibiotics, particularly amoxicillin, and the onset of cutaneous manifestations in a patient with nasopharyngeal carcinoma in terms of timing, rash morphology, clinical findings and triggering factors. The current evidence of this phenomenon is exclusively found in the pediatric population and in patients affected by acute EBV infection, with the absence of evidence in the adult population and in patients with EBV reactivation. Hence, researchers should be open to the possibility of investigating the genesis of these phenomena even in a population that is, in fact, antithetical to the one that is currently considered.

In conclusion, particular caution should be taken when prescribing antibiotics, particularly amoxicillin, in patients with EBV-related nasopharyngeal carcinoma.

## Figures and Tables

**Figure 1 viruses-17-00368-f001:**
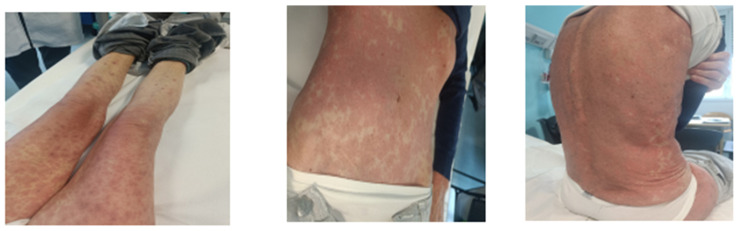
Morphology and distribution of the rash (images taken during the clinical examination).

**Figure 2 viruses-17-00368-f002:**
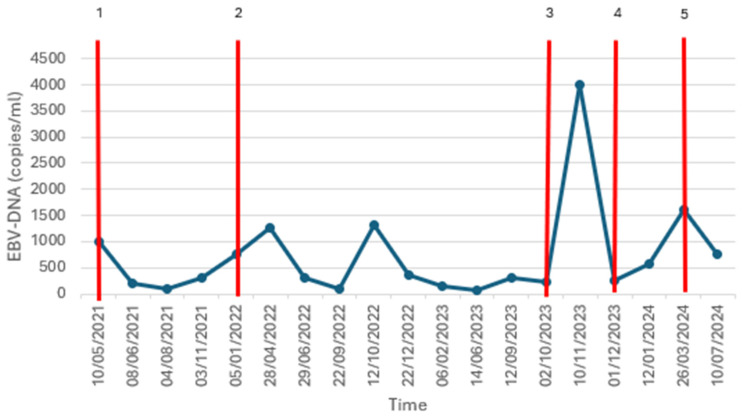
Correlation between time and EBV-DNA titer. Index: 1 = start of carboplatin–gemcitabin; 2 = start of pembrolizumab–olparib; 3 = onset of the rash; 4 = remission of the rash; 5 = start of paclitaxel.

## Data Availability

Data are contained within the article.

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
