# Peer review of "Amoxicillin-Induced Atypical Exanthema in a Patient with EBV-Related Nasopharyngeal Carcinoma: A Case Report"

_viruses, 2025, doi:10.3390/v17030368_

Round 1
Reviewer 1 Report
Comments and Suggestions for Authors
This case report is the first to describe a patient with EB virus (EBV)-related nasopharyngeal carcinoma who developed atypical rash after receiving amoxicillin treatment. It also explores the potential association between this skin reaction and the use of amoxicillin. This case fills a gap in the relevant field. By analyzing the morphology of the rash, its temporal relationship, and the changes in EBV-DNA load, the authors proposed a possible causal relationship between amoxicillin and the rash, providing a reference for clinicians. However, the article lacks innovation and sufficient data support, and has the following issues.
Major problems:
- Limited sample size. The study is based on a single case, which may limit the generalizability and reliability of the conclusions.
- Lack of a control group. The absence of a control group (e.g., EBV-related nasopharyngeal carcinoma patients not treated with amoxicillin) makes it difficult to further validate the causal relationship. Including such a group could provide stronger evidence.
- Lack of long-term follow-up data. The authors are advised to supplement the report with long-term follow-up data, including changes in EBV-DNA load and disease recurrence, to provide a more comprehensive assessment of the patient's prognosis.
- Lack of drug concentration monitoring. The authors should consider adding data on the blood concentration of amoxicillin in the patient to rule out the possibility that the rash was caused by drug overdose or abnormal metabolism.
- Lack of discussion on mechanisms. Although the study observed a significant increase in EBV-DNA load during the rash, the underlying mechanisms were not explored. The authors are encouraged to discuss potential biological mechanisms based on existing literature to explain this phenomenon.
Minor problems:
- Simplify the language. The authors should simplify the language to make the article clearer and more accessible.
- Update the reference. It is recommended to include some of the latest relevant research articles to ensure the timeliness of the content.
- Expand the discussion. The authors are advised to further explore the potential risks of other β-lactam antibiotics in similar situations in the discussion section and provide an analysis.
- It is recommended to cite the following relevant references to enhance the scientific rigor of the article.
- Analysis of the Results of Cytomegalovirus Testing Combined with Genetic Testing in Children with Congenital Hearing Loss.
- The Pathogenesis of Cytomegalovirus and Other Viruses Associated with Hearing Loss: Recent Updates.
- Cellular Processes Induced by HSV-1 Infections in Vestibular Neuritis.
- The Effects of Viral Infections on the Molecular and Signaling Pathways Involved in the Development of the PAOs.
Simplify the language of the article to make the sentences smoother and easier to understand.
Author Response
“Amoxicillin-Induced atypical exanthema in a patient with EBV-related nasopharyngeal carcinoma: a case report”
Response to Reviewer 1 Comments
|
||
1. Summary |
|
|
Thank you very much for taking the time to review this manuscript. Please find the detailed responses below and the corresponding revisions/corrections in track changes in the re-submitted files.
|
||
2. Questions for General Evaluation |
Reviewer’s Evaluation |
Response and Revisions |
Does the introduction provide sufficient background and include all relevant references? |
Can be improved |
|
Are all the cited references relevant to the research? |
Can be improved |
|
Is the research design appropriate? |
Can be improved |
|
Are the methods adequately described? |
Can be improved |
|
Are the results clearly presented? |
Can be improved |
|
Are the conclusions supported by the results?
|
Can be improved |
|
3. Point-by-point response to Comments and Suggestions for Authors |
||
Comments 1: Limited sample size. The study is based on a single case, which may limit the generalizability and reliability of the conclusions.
Response 1: Thank you for your comment. The paper is based on a single case report and is intended to suggest a possible correlation between amoxicillin intake and the onset of a rash in a patient with nasopharyngeal carcinoma. These findings should be further explored through larger and more generalizable studies.
|
||
Comments 2: Lack of a control group. The absence of a control group (e.g., EBV-related nasopharyngeal carcinoma patients not treated with amoxicillin) makes it difficult to further validate the causal relationship. Including such a group could provide stronger evidence.
|
||
Response 2: Thank you for your comment. The paper is based on a single case report and is intended to suggest a possible correlation between amoxicillin intake and the onset of a rash in a patient with nasopharyngeal carcinoma. These findings should be further explored through larger and more generalizable studies.
Comments 3: Lack of long-term follow-up data. The authors are advised to supplement the report with long-term follow-up data, including changes in EBV-DNA load and disease recurrence, to provide a more comprehensive assessment of the patient's prognosis.
Response 3: Thank you for your comment. Regarding the long term follow up: the patient progressed in March 2024 and started a third line systemic treatment with paclitaxel q21 days; at the moment the patient is still alive and in active treatment. We corrected the table and added the latest EBV-DNA assessment and line of treatment. (Line 98-105; Graphic 1)
Comments 4: Lack of drug concentration monitoring. The authors should consider adding data on the blood concentration of amoxicillin in the patient to rule out the possibility that the rash was caused by drug overdose or abnormal metabolism.
Response 4: Thank you for your comment. It was not possible to assess the antibiotic concentration in the patient's blood during and after the rash, as she was treated according to clinical practice, and these tests are not routinely performed. However, we confirm that the patient did not exceed the maximum recommended doses of the drug, as indicated for the condition it was administered for. Additionally, the patient had previously taken this medication before cancer diagnosis, without experiencing any adverse effects that would suggest an abnormal metabolism. We added a paragraph to clarify this aspect. (Line 135-136; Line 166-170)
Comments 5: Lack of discussion on mechanisms. Although the study observed a significant increase in EBV-DNA load during the rash, the underlying mechanisms were not explored. The authors are encouraged to discuss potential biological mechanisms based on existing literature to explain this phenomenon.
Response 5: Thank you for your comment. As stated in the discussion we believe that the exanthema was most likely linked with amoxicillin intake, based on the timing, morphology and symptoms of the rash and their consistency with the available evidence. The possible other aetiologies were ruled out.
Comments 6: Simplify the language. The authors should simplify the language to make the article clearer and more accessible
Response 6: Language was simplified.
Comments 7: Update the reference. It is recommended to include some of the latest relevant research articles to ensure the timeliness of the content.
Response 7: Thank you for your comment. We cited all the available evidence on this topic.
Comments 8: Expand the discussion. The authors are advised to further explore the potential risks of other β-lactam antibiotics in similar situations in the discussion section and provide an analysis.
Response 8: Thank you for your comment. We cited all the available evidence on this topic.
|

Reviewer 2 Report
Comments and Suggestions for Authors
I have the following concerns:
- a. Please provide the last follow-up regarding the course of the nasopharyngeal carcinoma. Is it in remission? b. Please provide a table mentioning all the therapies applied for the carcinoma, from diagnosis until today. c. How do you treat relapse in such cases?
- I was wondering if the underlying neoplasm, even in remission, at the time of appearance of the exanthema is linked pathophysiologically with the exanthema, due to immunosuppression for example, or other B or T cellular defects. Please analyze that to a separate paragraph in the discussion.
- Had the patient received the same (amoxicillin) or other antibiotics in the past and which were they? If the patient received amoxicillin in the past (in parallel with the carcinoma), then why did not she have the same exanthema?
- In the end, what triggered the exanthema and caused its appearance?
Author Response
“Amoxicillin-Induced atypical exanthema in a patient with EBV-related nasopharyngeal carcinoma: a case report”
Response to Reviewer 2 Comments
|
||
1. Summary |
|
|
Thank you very much for taking the time to review this manuscript. Please find the detailed responses below and the corresponding revisions/corrections highlighted/in track changes in the re-submitted files.
|
||
2. Questions for General Evaluation |
Reviewer’s Evaluation |
Response and Revisions |
Does the introduction provide sufficient background and include all relevant references? |
Yes |
|
Are all the cited references relevant to the research? |
Yes |
|
Is the research design appropriate? |
Yes |
|
Are the methods adequately described? |
Yes |
|
Are the results clearly presented? |
Yes |
|
Are the conclusions supported by the results?
|
Yes |
|
3. Point-by-point response to Comments and Suggestions for Authors |
||
Response 1: Thank you for your comment. A) The disease is not in remission, the patient progressed in March 2024 and started a second line systemic treatment with weekly paclitaxel; at the moment the patient is still alive and in active treatment. B) We corrected the table and added the latest EBV-DNA assessment and line of treatment. C) The patient is not in remission. (Line 98-105; Graphic 1)
|
||
Comments 2: I was wondering if the underlying neoplasm, even in remission, at the time of appearance of the exanthema is linked pathophysiologically with the exanthema, due to immunosuppression for example, or other B or T cellular defects. Please analyze that to a separate paragraph in the discussion.
|
||
Response 2: Thank you for your comment. During the disease the patient was not immunosuppressed. Immunotherapy is not an immunosuppressive therapy and is not associated with a decline in T cell and B cell levels. We added a paragraph specifying this aspect. (Line 160-164)
Comments 3: Had the patient received the same (amoxicillin) or other antibiotics in the past and which were they? If the patient received amoxicillin in the past (in parallel with the carcinoma), then why did not she have the same exanthema?
Response 3: Thank you for your comment. The patient received a previous therapy with amoxicillin before the diagnosis of nasopharyngeal cancer. To the best of our knowledge the patient didn’t receive any antibiotic therapy with amoxicillin thereafter. We added a paragraph specifying this aspect. (Line 75; Line 166-170)
Comments 4: In the end, what triggered the exanthema and caused its appearance?
Response 4: Thank you for your comment. As stated in the discussion we believe that the exanthema was most likely linked with amoxicillin intake, based on the timing, morphology and symptoms of the rash and their consistency with the available evidence. The possible other aetiologies were ruled out.
|
Round 2
Reviewer 1 Report
Comments and Suggestions for Authors
The authors report a case of atypical exanthema associated with amoxicillin intake, exploring the potential risks of using beta-lactam antibiotics in patients with EBV-related nasopharyngeal carcinoma. This case provides a reference for clinicians treating such patients. The authors have already provided detailed answers to the questions I raised.
Reviewer 2 Report
Comments and Suggestions for Authors
I have no further concerns.